# Comparative GWAS using global and Indian Reference Panels reveals non-coding drivers of COVID-19 severity and mortality

Aastha Kaushik[1‡], Ramakant Mohite[1‡], Ranjeet Maurya[1,2], Bansidhar Tarai[3], Sandeep Budhiraja[3], Uzma Shamim[1*], Rajesh Pandey[1,2*]

1 Division of Infectious Disease Biology, INtegrative GENomics of HOst-PathogEn (INGEN-HOPE) laboratory, CSIR-Institute of Genomics and Integrative Biology (CSIR-IGIB), Delhi, India, 2 Academy of Scientific and Innovative Research (AcSIR), Ghaziabad, India, 3 Max Super Speciality Hospital (A Unit of Devki Devi Foundation), Max Healthcare, Delhi, India

‡ These authors are joint senior authors on this work.
* rajeshp@igib.in, rajesp.igib@csir.res.in (RP); uzma.shamim@igib.in (US)

## Abstract

India remains underrepresented in global genomic studies. We hypothesized that population-specific genetic variants contribute to COVID-19 severity and outcomes, and that the choice of reference panel during imputation impacts Genome-Wide Association Studies (GWAS) resolution. Integrating both global and indigenous reference panels may unravel unique and shared genetic associations that are otherwise missed during standard analyses. In this study, we aimed to perform a comparative GWAS using Indian population-specific (IndiGen) and global (1000 Genomes Project/1KGenomes) reference panels to identify potential genetic loci associated with the COVID-19 differential severity and mortality among the Indian patients. Genomic DNA was extracted and genotyped from the patients who were stratified based on the clinical data capturing COVID-19 symptoms and clinical outcomes. Quality control, liftover, phasing and imputation were performed on the genotypic data. GWAS was performed separately for the severity and mortality phenotypes. Significant loci were functionally annotated using Linkage Disequilibrium (LD) analysis, eQTL mapping, and gene annotation tools. Comparative GWAS with 1KGenomes and IndiGen panels revealed both shared and unique loci. 1KGenomes identified protective variants near *MIR4432HG* involved in endothelial stability, while IndiGen uncovered risk variants with rs10096505 (*SFTPC/BMP1*) linked to alveolar collapse and fibrotic remodelling. rs9547631 was common to both panels for mortality, whereas IndiGen-specific risk variants (rs78554880, rs112982286, rs111390553, and rs79900659) were associated with immune dysregulation. Functional annotation of these loci pointed to key biologically plausible links to COVID-19 severity and fatal outcomes. Briefly, the use of an indigenous reference panel improved variant discovery and LD resolution, highlighting that population-specific signals are missed by the generic global datasets.

**Data availability statement:** The data supporting the findings of this study have been deposited in Figshare under the DOI: [https://doi.org/10.6084/m9.figshare.29650937] and are available under a CC-BY license.

**Funding:** This study received financial support from Bill and Melinda Gates Foundation (BMGF), (Grant number - INV-033578) awarded to RP. The funders had no role in study design, data collection and analysis, decision to publish, or preparation of the manuscript.

**Competing interests:** The authors have declared that no competing interests exist.

Our findings underscore the importance of inclusive genomic resources for accurate association mapping in the underrepresented populations.

## Author summary

Most genetic studies have focused on people of European descent, leaving South Asian populations, especially those from India, largely underrepresented. To help fill this gap, we studied the genetic makeup of Indian individuals with different COVID-19 severity levels and outcomes, ranging from recovery to death. We wanted to understand why some people become severely ill and while others recover, and whether genetic differences might help explain this. To study this, we analysed each person's DNA and used two different datasets to fill in missing genetic information. First, represents global populations (1KGenomes), and secondly, IndiGen, which is specific for the Indian population. The Indian specific dataset helped us discover more genetic differences, including some that were missed by the global reference. These differences were linked to important biological processes such as lung function and immune response. For instance, we identified a variant located near the *SFTPC* and *BMP1* genes, which are associated with impaired surfactant production and lung fibrosis. In patients who did not survive, we saw strong genetic signals associated with immune system regulation. In a nutshell, our study captures novel genetic signals with potential links to COVID-19 pathophysiology and highlights the importance of using tailored genomic resources to improve the accuracy of association findings.

## Introduction

The COVID-19 pandemic, caused by the SARS-CoV-2 virus, has had a devastating and uneven impact on global public health. Clinical outcomes range from asymptomatic infection to severe respiratory failure and multi-organ dysfunction [1,2]. Mortality rates varied significantly across regions and time periods [3,4]. Demographic factors, such as age, sex, and comorbidities (e.g., cardiovascular diseases, diabetes, and obesity), are well-established predictors of COVID-19 outcomes. Beyond these demographic and clinical predictors, increasing evidence highlights host genetic factors as important modulators of disease susceptibility, severity and outcome [5–7].

To investigate the genetic architecture underlying COVID-19 outcomes, several large-scale genome-wide association studies (GWAS) have been conducted globally through international collaborations such as the COVID-19 Host Genetics Initiative, the Severe COVID GWAS Group, GenOMICC, 23andMe, and AncestryDNA [8–11]. These efforts have led to the identification of multiple loci linked to COVID-19 risk, including 3p21.31 (*LZTFL1*) and 9q34.2 (*ABO*), which are linked to disease susceptibility and severity. However, most of these findings are derived from individuals of European ancestry, limiting their generalizability to other populations, especially

South Asian (SAS), a group that remains significantly underrepresented in global genetic studies, even today, and that's true for most of the health challenges.

A limited number of studies have explored COVID-19 outcomes in non-European populations, including Chinese and Middle Eastern cohorts [12,13]. These studies identified ancestry-specific associations and emphasized the importance of population-specific genetic variation in shaping immune response and viral pathogenesis. In India, a genetically diverse country with a high COVID-19 burden, genomic studies of COVID-19 host genetics remain limited. One such study by Prakrithi et al. applied polygenic risk scores (PRS) derived from international GWAS data to Indian subpopulations, using the Indian Genome Variation Consortium (IGVC) dataset [14]. While informative, the study relied on external summary statistics and did not involve direct genotyping of Indian COVID-19 patients, thereby limiting its resolution and applicability. Other Indian studies have focused on the candidate gene associations, such as those involving *TMPRSS2, EGLN1,* and *IL-10*, which showed correlations with COVID-19 severity in specific Indian subpopulations [15–17]. However, these efforts were constrained by small sample sizes, restricted ethnic representation, and a narrow candidate gene focus, lacking genome-wide coverage and functional validation.

India has a complex genetic landscape, shaped by historical migration, geographic isolation, and extensive population substructure, which makes it imperative to conduct ancestry-aware GWAS in this population [18,19]. Subgroups defined by ethnic, caste, linguistic, and regional affiliations possess distinct genomic backgrounds [20]. If unaccounted for, they can result in spurious GWAS signals due to confounding by ancestry rather than true disease associations. Population structure has been shown to influence genomic associations with COVID-19 severity. For instance, a study by Pandit et al. reported associations between polymorphic loci around genes involved in lung and heart function *(TNFSF4, TNFSF18, GOT2P2, LRRC74A),* the innate immune system *(DHX15),* and mitochondrial regulation *(PPARGC1A)* with increased severity of COVID-19 in the Western Indian cohort. Variants near *RFX3* and *UBXN10* were reported to be protective [21]. Although biologically informative, this study did not evaluate imputation performance in the Indian-specific context. Its findings, although insightful, highlight the need for comprehensive, ancestry-aware GWAS with rigorous imputation and stratification controls to validate and expand on such associations in diverse Indian populations. In addition, strong founder effects in certain small groups give rise to genetically homogeneous populations, which can inflate or mask associations by over-representing rare alleles and introducing bias into the genomic inference [22]. Another important limitation in previous GWAS is use of non-representative imputation panels for genotype imputation, particularly in genetically diverse populations [23].

Genotype imputation is a critical step in GWAS for enhancing variant coverage and statistical power, but its accuracy is highly dependent on the reference panel used. The widely used 1000 Genomes Project Phase III panel (1KGenomes), while globally representative, includes limited SAS population representation [24], which may hinder accurate imputation and LD resolution in the Indian cohorts. As a result, important population-specific genetic associations may be missed.

To address these challenges, we employed a dual reference imputation strategy. We used both the 1KGenomes reference panel and the IndiGen reference panel, developed under the pilot phase of the 'IndiGen' program and representing Indian-specific genomic variation [25]. While the 1KGenomes panel enables global comparability, the IndiGen reference panel provides LD patterns and ancestry-specific allele frequencies relevant to the Indian population, which are critical for imputation accuracy in the Indian datasets. To our knowledge, no published COVID-19 GWAS has utilized the IndiGen as a reference panel for imputation, making this study the first of its kind.

Our study aimed to bridge this gap by performing a genotype-based GWAS of COVID-19 severity and outcomes in an Indian population. We stratified patients into mild, moderate and severe categories based on the clinical criteria and analyzed them for severity and outcome-based associations. By employing both global and Indian reference panels for the imputation, we assessed the comparative performance of these panels in SNP detection and LD resolution; identified both shared (generalizable) and population-specific loci linked to the COVID-19 severity; functionally annotated lead variants using eQTL databases and gene expression profiles, and demonstrated the critical role of reference panel choice in

detecting true signals in GWAS conducted in diverse populations. Through this comprehensive design, our study contributes to the understanding of COVID-19 host genetics in India and provides a framework for methodologically sound and population-relevant genomic research in other infectious diseases, like Dengue, where ancestry aware genetic analyses may similarly enhance discovery

## Results

### Quality review and analysis

Following stringent QC procedures and removal of the outlier samples, a final dataset comprising 608 unrelated individuals was retained for further analysis. Post-QC, 729,224 high-quality variants were retained in the imputed dataset using the 1KGenomes Project reference panel, while the IndiGen reference panel imputation yielded 1,133,717 variants, reflecting a higher variant density likely due to its population-specific relevance. PCA analysis revealed that the study cohort clustered predominantly with the South Asian (SAS) super-population (Fig 1B**-i).** In the PCA with 1000 Genomes reference populations, the first two PCs explained 53.34% (PC1) and 25.42% (PC2) of the total genetic variance, respectively. A modest overlap toward the EAS cluster was observed for a subset of individuals, which is consistent with known historical admixture and regional genetic continuity within the Indian populations. These samples did not represent distinct population outliers and were therefore retained. In the within-cohort PCA, PC1 and PC2 explained 16.82% and 9.10% of the total variance, respectively. Two clear outliers were identified in this analysis and removed before imputation and downstream analyses.

### Clinical characteristics of the patients in the final cohort

The final study cohort included 608 SARS-CoV-2-infected unrelated individuals, categorized into control (*Mild*, N = 253), case (*Moderate + Severe*, N = 355), *recovered* (N = 524), and *deceased* (N = 66) groups according to the disease severity and clinical outcome. Age distribution followed the expected trend, with older individuals more frequent in the severe and deceased groups (mean age: 51.1 years in controls, 60.2 years in cases, and 65.5 years in the deceased group). Clinical observations at the time of admission reflected severity-dependent respiratory decline. Mean peripheral oxygen saturation dropped from 97.27% in controls to 87.41% in the deceased patients. Respiratory rate followed an opposite trajectory, increasing from 20.07 to 23.18 breaths per minute between the same groups. These physiological patterns are consistent with the impaired pulmonary function in more severe disease. Male patients were more prevalent across all the groups, with the proportion highest among the case (76.6%) and deceased (76.9%) subgroups. Female representation ranged from 31.6% in the control (mild) group to 23.1% among the non-survivors. These baseline features reflect the clinical trajectory of COVID-19 across severity categories and are showcased in **Table 1** and **Fig 1A**.

### IndiGen outperforms 1KGenomes in detecting and classifying COVID-19-associated variants

We performed genome-wide analyses using imputed genotype data from the two reference panels, 1KGenomes and IndiGen, across the two phenotype comparisons: *Control (Mild) vs Case (Moderate + Severe)* and '*Recovered vs Deceased*' groups. Quality-controlled genotype data were analyzed under an additive genetic model, with age, sex, population structure (PC1-PC10), and comorbidities included as covariates. While no variant surpassed the conventional genome-wide significance threshold ($p < 5 \times 10^{-8}$), several suggestive associations ($p < 1 \times 10^{-5}$) with biologically plausible effects were detected (**Table 2**). The corresponding Manhattan and QQ plots for each panel and comparison group are shown in **Fig 2**, demonstrating the genomic distribution and inflation of the test statistics. To compare variant detection differences between the two imputation panels and phenotypic groups, the distribution of SNPs using a lollipop plot and a heatmap was visualized and is illustrated in **Fig 1C**. Overall, the IndiGen panel identified a greater number of variants than the 1KGenomes panel. This difference was more evident in the '*Recovered* vs *Deceased*' group, suggesting better resolution

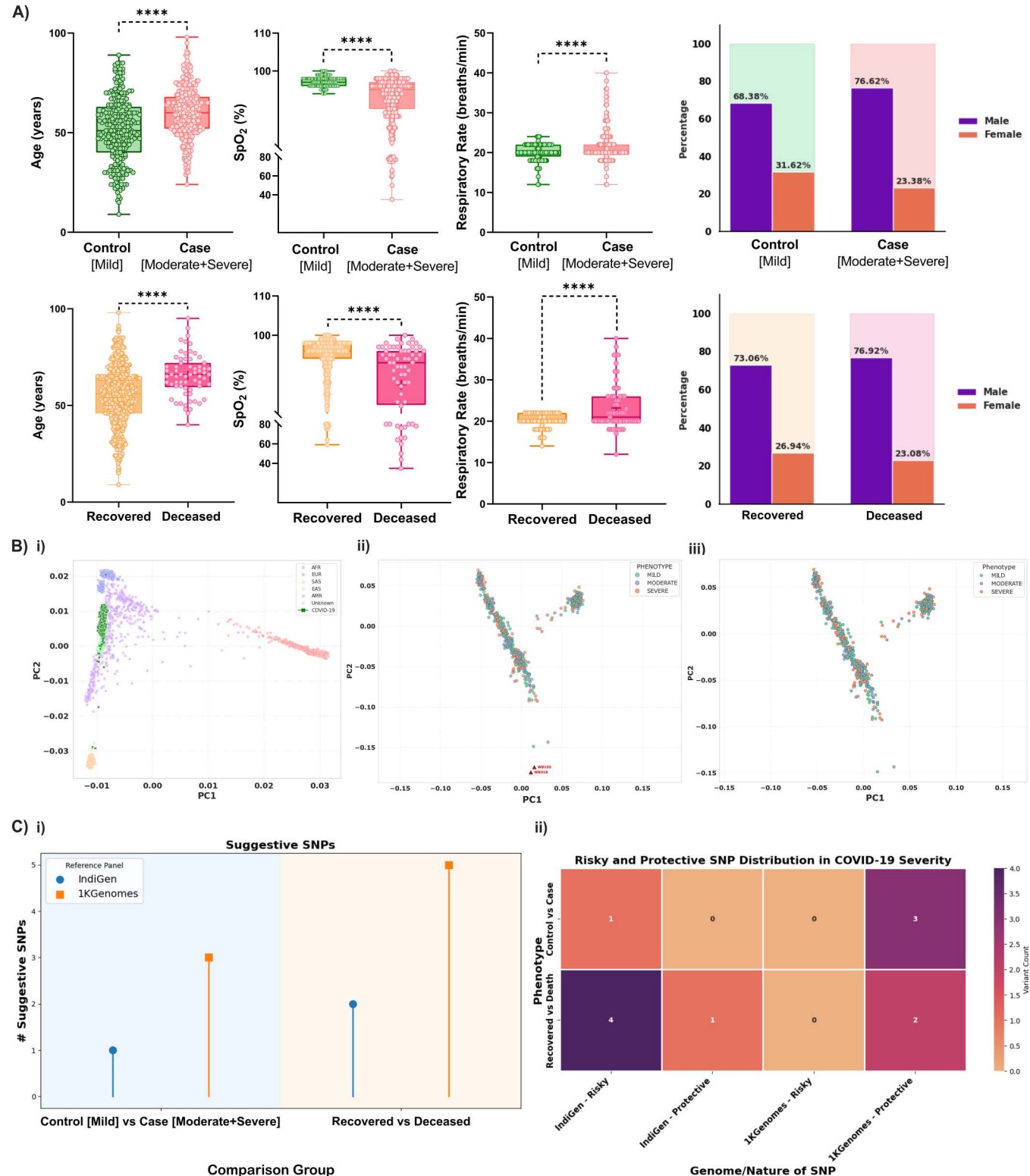

**Fig 1. Clinical characteristics, Principal Component Analysis (PCA), and summary of GWAS findings in COVID-19. A)** Boxplots displaying clinical parameters (Age, SpO$_2$, Respiratory Rate and percentage of male and female patients) across phenotypic groups: '*Control (Mild) vs Case (Moderate + Severe)*' (top row) and '*Recovered vs Deceased*' (bottom row). All comparisons were made using the t-test in GraphPad Prism v8.0.2 (licensed

version), and are statistically significant (p-value<0.0001); **B)** PCA of COVID-19 genotyped samples: **i)** Ancestry PCA showing clustering with the South Asian (SAS) population and partial overlap with the East Asian (EAS), confirming expected ancestry; **ii)** PCA before outlier removal: Two outliers were found; **iii)** PCA after outlier removal: Outliers were removed, illustrating more homogeneous clustering across phenotypic groups; **C)** Summary of GWAS findings in COVID-19: **i)** Lollipop plot comparing the number of suggestive SNPs identified in two phenotype groups '*Control (Mild) vs Case (Moderate + Severe)*'and '*Recovered vs Deceased*' across the two reference panels (1KGenomes and IndiGen); **ii)** Heat map illustrating the distribution and classification (risk or protective) of the suggestive SNPs across the phenotype subgroups and reference panels, highlighting differences in the variant contribution to the COVID-19 severity and outcome.

**Table 1.  Summary of clinical characteristics of patients in the study.**

| Characteristics | Control (*Mild*) | Case (*Moderate + Severe*) | Recovered | Deceased |
|---|---|---|---|---|
| Number of patients | N=253 | N=355 | N=524 | N=66 |
| Mean Age (years) | 51.11 | 60.21 | 55.13 | 65.51 |
| Mean SpO2 (%) | 97.27 | 92.35 | 95 | 87.41 |
| Mean Respiration rate (RR/min) | 20.07 | 21.71 | 20.04 | 23.18 |
| Male (%) | 68.38 | 76.62 | 73.06 | 76.92 |
| Female (%) | 31.62 | 23.38 | 26.94 | 23.08 |

**Table 2.  Summary of top suggestive SNPs associated with COVID-19 severity and disease outcomes in the 1KGenomes and IndiGen imputed data.**

| Reference panel | Group | SNP | CHR | BP | REF | ALT | A1 | TEST | Freq_Case | Freq_Control | OR | STAT | P-value | Nature of SNP |
|---|---|---|---|---|---|---|---|---|---|---|---|---|---|---|
| 1KGenomes | *Control (Mild) vs Case (Moderate + Severe)* | rs35575084 | 2 | 60307089 | T | C | C | ADD | 0.2451 | 0.3636 | 0.5369 | -4.537 | 5.72E-06 | Protective |
| 1KGenomes | *Control (Mild) vs Case (Moderate + Severe)* | rs17024964 | 2 | 60307203 | C | T | T | ADD | 0.2451 | 0.3656 | 0.5307 | -4.612 | 3.99E-06 | Protective |
| 1KGenomes | *Control (Mild) vs Case (Moderate + Severe)* | rs34607367 | 2 | 60307320 | A | G | G | ADD | 0.2451 | 0.3617 | 0.5395 | -4.485 | 7.31E-06 | Protective |
| IndiGen | *Control (Mild) vs Case (Moderate + Severe)* | rs10096505 | 8 | 22167010 | G | A | A | ADD | 0.3127 | 0.3893 | 1.793 | 4.509 | 6.50E-06 | Risky |
| 1KGenomes | *Recovered vs Deceased* | rs9547631 | 13 | 36650644 | T | C | C | ADD | 0.3042 | 0.3794 | 0.294 | -4.681 | 2.86E-06 | Protective |
| 1KGenomes | *Recovered vs Deceased* | rs1850535 | 13 | 36655911 | T | C | C | ADD | 0.4634 | 0.3419 | 0.3076 | -4.519 | 6.22E-06 | Protective |
| IndiGen | *Recovered vs Deceased* | rs78554880 | 2 | 739515 | G | A | A | ADD | 0.03521 | 0.04348 | 4.845 | 4.575 | 4.75E-06 | Risky |
| IndiGen | *Recovered vs Deceased* | rs112982286 | 2 | 743200 | AG | A | A | ADD | 0.03521 | 0.04348 | 4.845 | 4.575 | 4.75E-06 | Risky |
| IndiGen | *Recovered vs Deceased* | rs111390553 | 2 | 743856 | A | G | G | ADD | 0.03521 | 0.04348 | 4.845 | 4.575 | 4.75E-06 | Risky |
| IndiGen | *Recovered vs Deceased* | rs79900659 | 2 | 745846 | T | C | C | ADD | 0.03521 | 0.04348 | 4.845 | 4.575 | 4.75E-06 | Risky |
| IndiGen | *Recovered vs Deceased* | rs9547631 | 13 | 36650644 | T | C | C | ADD | 0.3127 | 0.3893 | 0.2889 | -4.748 | 2.06E-06 | Protective |

*# **CHR:** Chromosome; **BP:** Chromosomal position in base pair; **REF:** Reference allele; **ALT:** Alternate allele; **A1:** Affected Allele; **TEST:** Type of genetic association model used (ADD=Additive model); **Freq_Case**: Frequency of effect allele in cases; **Freq_Control**: Frequency of effect allele in controls; **OR:** Odd's Ratio; **STAT:** Test statistic value (Z-Score).*

captured for outcome-specific association (Fig 1C-**i**). We further classified these variants based on their effect direction. Risk-associated variants were more frequently observed in the IndiGen panel, particularly in the deceased group, whereas the majority of SNPs captured by the 1KGenomes panel were classified as 'protective' (Fig 1C-**ii**). These results demonstrate clearly that the choice of reference panels influences the number of variants detected, but also the inferred direction in association testing.

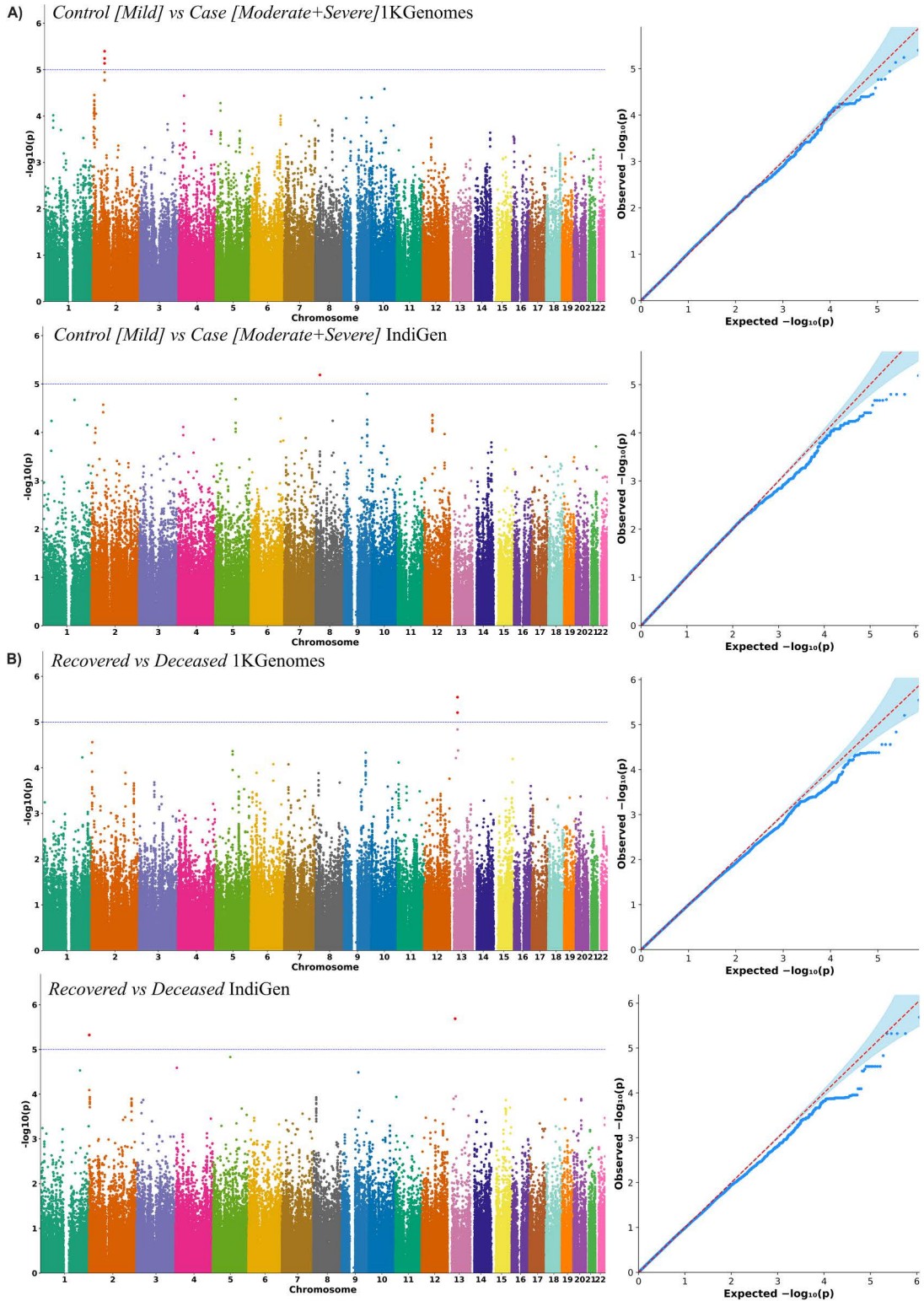

**Fig 2. GWAS results for COVID-19 severity and mortality using 1KGenomes and IndiGen-imputed datasets. A)** Results from the '*Control (Mild) vs Case (Moderate + Severe)*' comparison group using 1KGenomes and IndiGen-imputed data, showing Manhattan plots (right) and corresponding QQ plots (left); **B)** Results from the '*Recovered vs Deceased*' group, with Manhattan plots (right) and QQ plots (left). In each Manhattan plot, the -log$_{10}$

(*p-values*) of SNP associations are plotted against the genomic position, with alternating chromosome colours. The horizontal line marks the suggestive significance threshold (p-value = 1 × 10⁻⁵). The QQ plots depict observed versus expected *p-value* distributions, with deviation from the diagonal red line indicating potential true associations. Notably, both reference panels showed signs of potential genetic associations, with some phenotypes displaying more significant and clustered signals, indicating stronger links to the COVID-19 outcomes.

## Distinct genetic associations revealed by 1KGenomes and IndiGen

In the 1KGeomes-imputed COVID-19 dataset, a distinct signal emerged on chromosome 2, where three closely spaced (within a ~ 200 bp region) SNPs (rs35575084, rs17024964, and rs34607367) were significantly associated, exhibiting p-values below 7.5 × 10⁻⁶. Each of these SNPs showed lower allele frequencies in the case group relative to the controls, suggesting a protective effect, which was confirmed after covariate adjustment during the logistic regression analysis. These associations highlight a potential regulatory hotspot warranting further investigation. In contrast, the IndiGen panel uniquely identified rs10096505 on chromosome 8 as a suggestive variant (p-value = 6.5 × 10⁻⁶) associated with the severity, a signal absent in the 1KGenomes panel, highlighting the value of population-specific reference panels. Although the alternate allele (A) of rs10096505 was less frequent in the case group (31.3%) compared to the controls (38.9%), covariate-adjusted logistic regression revealed it to be a risk allele (Odds Ratio, OR=1.79), suggesting potential confounding effects that were accounted for in the model. This highlights the importance of adjusted analyses in uncovering the true genetic contribution to the disease severity. For the '*Recovered vs Deceased*' comparison, rs9547631 on chromosome 13 emerged as a shared signal across both panels, suggesting a consistent association with mortality risk. Additionally, the IndiGen panel identified a robust multi-SNP signal on chromosome 2 spanning four variants (rs78554880, rs112982286, rs111390553 and rs79900659), all significantly associated with mortality (p-value < 4.8 × 10⁻⁶) and exhibiting OR > 4.8. Despite nearly identical raw allele frequencies between groups, these variants revealed significance only after adjustment for covariates, underscoring the importance of multivariate modelling, particularly for the low-frequency alleles. These findings formed the basis for the downstream LD mapping and functional annotation in the downstream analysis, providing candidate loci for further biological validation.

## Chr2 lincRNA/miRNA locus is associated with endothelial pathology in severe COVID-19

Distinct suggestive loci and their surrounding LD patterns were identified in both panels, which are summarized in S2 Table, while the SNPs showcasing strong LD and their genomic coordinates, with the suggestive SNPs, are highlighted in **Fig 3**. Using the 1KGenomes reference panel, in the '*Control (Mild) vs Case (Moderate + Severe)*' comparison, a prominent LD block was observed on chromosome 2 around the suggestive SNP rs35575084 (chr2:60307089, p-value = 5.72E-06, OR=0.536), which formed a high confidence LD block (r² > 0.93) with rs6545801, rs6545803, rs17024964, rs34607367, rs35998257, and rs35196779 (**Fig 3**, S2 Table). These SNPs are located near the long intergenic non-coding RNA gene (lincRNA) *AC007100.1* and near *MIR4432HG*, which is the host gene for *microRNA-4432* (**Fig 4**) (S3 Table). As annotated by SNPNexus, no direct overlap was found with the coding genes, which suggests that these SNPs may influence regulatory pathways through non-coding RNAs and raises the possibility of a novel locus linked to COVID-19 severity. Functional insights indicate that *miR-4432*, which is derived from *MIR4432HG*, directly inhibits *FGFBP1* in human endothelial cells [26]. *FGFBP1* is a secreted modulator of fibroblast growth factor activity, and it's an important regulator of endothelial dysfunction, and has also been described as a biomarker of prolonged SARS-CoV-2 replication in the ventilated COVID-19 patient [27]. Collectively, these observations provide a basis for a non-coding regulation that could affect endothelial dysfunction, which is a well-known feature of severe COVID-19 pathophysiology.

## IndiGen identified a variant at Chr8 as a potential modulator of fibrotic and surfactant pathways in COVID-19

In contrast, analysis using the Indian-specific IndiGen reference panel identified rs10096505 (chr8:22167010) as a suggestive SNP in the '*Control (Mild) vs Case (Moderate + Severe)*' association. This SNP showed no strong LD with

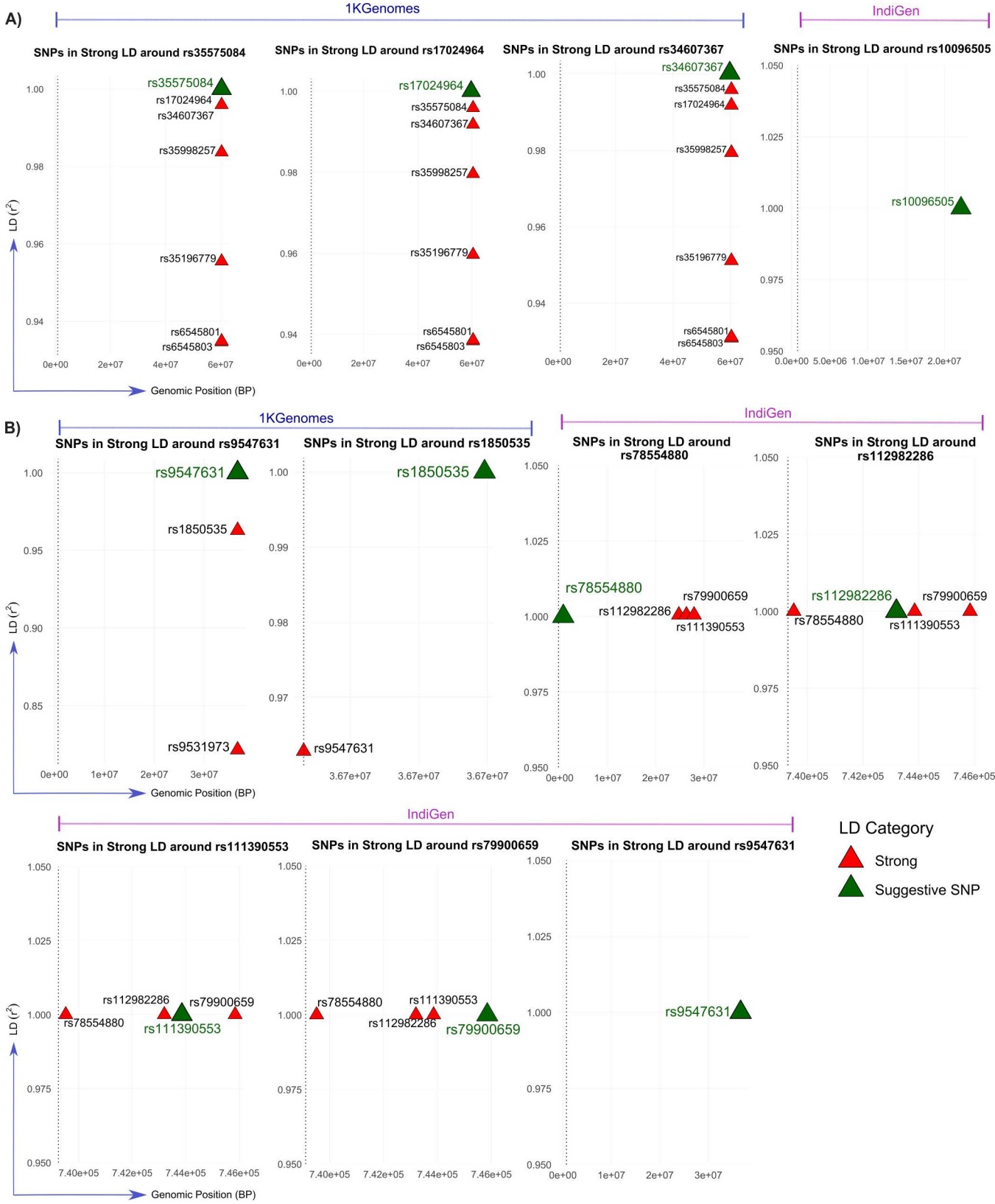

**Fig 3. Genomic landscape of SNPs in strong LD with suggestive SNPs associated with COVID-19 severity and outcome using 1KGenomes and IndiGen-imputed data.** This figure illustrates the regional plots of SNPs exhibiting strong LD ($r^2 \geq 0.8$) with suggestive SNPs identified from GWAS

in two phenotypic comparisons, **A)** *Control (Mild) vs Case (Moderate + Severe)*, and **B)** *Recovered vs Deceased*. Results are displayed separately for the 1KGenomes and IndiGen reference panels. Each sub-panel highlights the genomic region surrounding a suggestive SNP (green triangles), with nearby SNPs in strong LD (red triangles). The x-axis indicates the genomic position (base pair), and the y-axis represents the $-\log_{10}$(p-value) from the association tests. This figure highlights the added value of using a population-matched reference genome (IndiGen) in capturing more LD structure accurately, thereby improving GWAS resolution in underrepresented populations. These regional plots also reveal loci that may harbour causal variants influencing COVID-19 severity and survival, underscoring the importance of fine-mapping in diverse populations.

the nearby variants (S2 Table, **Fig 3**), suggesting an isolated signal potentially unique to the Indian population. However, functional annotation using GTEx revealed that rs10096505 is a strong eQTL for *BMP1* across multiple tissues, including blood, pancreas, thyroid, and brain, with (p ≤ $5.4 \times 10^{-11}$; S4 Table). *BMP1* encodes a secreted metalloprotease essential for the extracellular matrix remodelling, collagen processing, and activation of latent TGF-β, a major profibrotic cytokine [28,29]. In addition to its role as a *BMP1*eQTL, rs10096505 also acts as a lung-specific sQTL for *SFTPC* (Surfactant Protein C), with significant effects on splicing in the lung and brain tissues (NES ± 0.59; S4 Table). *SFTPC* is expressed in alveolar type II epithelial cells and is critical for surfactant homeostasis and alveolar stability [30,31]. Clinical studies show a marked decline in *SFTPC* expression in critically ill COVID-19 patients, implicating surfactant depletion in the alveolar collapse [32].

These findings suggest that genetic variation at rs10096505 may simultaneously regulate *BMP1*-mediated fibrotic signalling and *SFTPC*-dependent surfactant homeostasis, which are implicated in lung injury, fibrosis, and COVID-19 pathophysiology. Importantly, gene mapping across ±1Mbp of rs10096505 confirmed proximity to a cluster of immune and lung-expressed genes, including *BMP1 (bone morphogenetic protein 1,*Gene ID: 641), *SFTPC (surfactant protein C,* Gene ID: 6440), *HR (HR lysine demethylase and nuclear receptor corepressor), REEP4 (receptor accessory protein 4), NUDT18 (nudixhydrolase 18,* Gene ID: 79873), *PHYHIP (phytanoyl-CoA 2-hydroxylase interacting protein*, Gene ID: 9796) and *POLR3D (RNA polymerase III subunit D,* Gene ID: 661) (**Fig 4**) [33,34]. While not directly associated, their proximity to rs10096505 highlights the regulatory potential of this transcriptionally active region.

## Consistent Chr13 signal and novel Chr2 region linked with mortality due to COVID-19

We next compared recovered (N = 524) and deceased (N = 66) individuals to identify potential host genetic variants contributing specifically to COVID-19 mortality. Among the loci identified, a signal on chromosome 13 was consistently observed in both the 1KGenomes and IndiGen reference panels. Using the 1KGenomes reference panel for imputation, the lead suggestive variant, rs9547631 (chr13:36650644, p-value = 2.86E-06, OR=0.294), together with rs9531973 and rs1850535 (r² = 0.82–0.96), maps downstream of *SERTM1 (serine-rich and transmembrane domain-containing 1,* Gene ID: 400120) and upstream of the pseudogene *HIST1H2APS6 (H2AC histone family pseudogene family 1,* Gene ID: 100509927) (**Fig 5**) (S3 Table). Although no significant eQTLs were found in GTEx, *SERTM1* is localized in the intracellular membrane-bound organelles with a potential role in cell signalling. In the IndiGen panel, rs9547631 was again identified as a suggestive SNP, and its cross-population reproducibility of this association suggests that this locus may reflect a conserved signal relevant to COVID-19 mortality.

'Roadmap Epigenome Project' data, which maps chromatin features across different human tissues, was accessed via SNPNexus, and it revealed that rs9547631 lies in an open-chromatin region in brain tissue, suggesting a regulatory role in neuro-COVID-19 symptoms, and shows repressive marks in muscle, possibly linked to fatigue (S6 Table). The variant rs1850535 (p-value = 6.22E-06, OR=0.30) overlaps repressive chromatin in immune, cardiac, neural and gut tissues, indicating potential involvement in immune dysregulation and multi-organ effects seen in severe COVID-19 (S6 Table). A second novel association was observed on chromosome 2, exclusively in the India-specific IndiGen panel. Four perfectly linked variants (r² = 1.0), rs78554880 (p-value = 4.75e-06, OR=4.845), rs112982286 (p-value = 4.75e-06, OR=4.845),

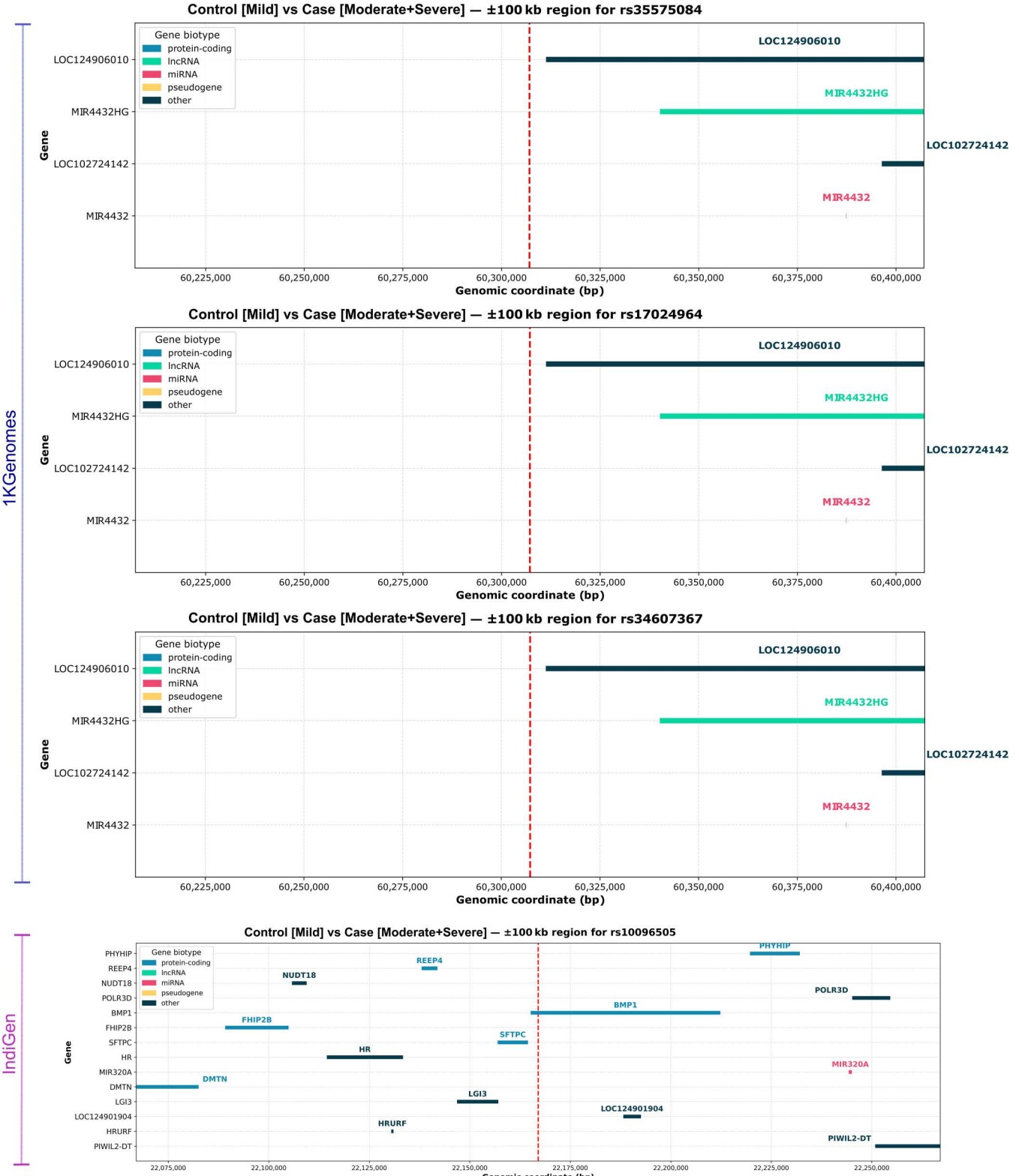

**Fig 4. Genomic landscape of suggestive SNPs in *Control (Mild) vs Case (Moderate + Severe)*.** Genomic context of suggestive variants associated with COVID-19 severity '*Control (Mild) vs Case (Moderate + Severe)*' from 1KGenomes and IndiGen panels: Genomic regions surrounding lead suggestive SNPs identified in the '*Control (Mild) vs Case (Moderate + Severe)*' GWAS are shown, highlighting the nearby genes and non-coding elements.

Panels (top to bottom) display rs35575084, rs17024964, and rs34607367 identified using the 1KGenomes panel, located near *MIR4432HG* and *MIR4432* lncRNA regions on chromosome 2; and rs10096505, a severity-associated suggestive SNP detected only with IndiGen-imputed data, located near *BMP1* and *SFTPC* on chromosome 8.

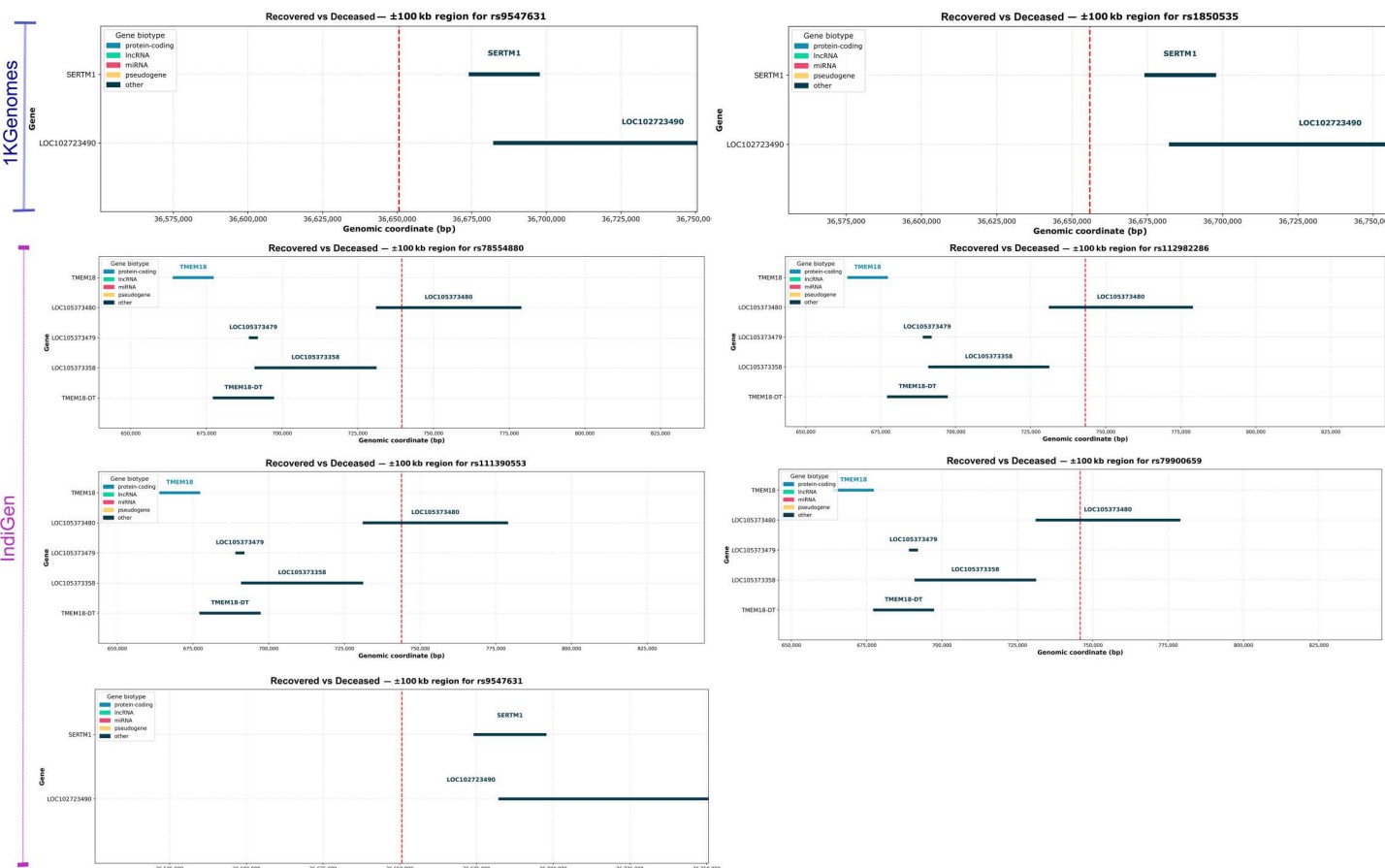

**Fig 5. Genomic landscape of suggestive SNPs in *Recovered vs Deceased*.** Genomic context of suggestive variants associated with COVID-19 outcomes ('*Recovered vs Deceased*') from 1KGenomes and IndiGen panels: Regional gene maps surrounding suggestive SNPs associated with mortality outcomes across reference panels are shown. Panels (top to bottom) showcase rs9547631 and rs1850535, which were identified using 1KGenomes, near *SERTM1* and *LOC102723049* on chromosome 13. rs78554880, rs112982286, rs111390553, and rs79900659 were uniquely detected in IndiGen-imputed data, clustered on chromosome 2 near lncRNAs such as *AC116609.1* and *LINC01115*. The rs9547631 was also replicated in the IndiGen-imputed data, reinforcing its potential as a cross-population mortality-associated variant. Each plot marks the SNP (red-dashed line) and visualises adjacent genes categorised by biotype. These findings highlight distinct regulatory loci identified through population-specific reference panels.

rs111390553 (p-value = 4.75e-06, OR=4.845), and rs79900659 (p-value = 4.75e-06, OR=4.845) (**Fig 3**, S2 Table), reside within or near *AC116609.1* and *AC116609.2*, both annotated as lincRNAs (S3 Table). Further exploration revealed that rs78554880 overlaps both repressive (H3K27me3) and active (H3Kme1, H3K4me3, H3K27ac) histone marks across a wide range of immune-related tissues, including CD4[+] T cells, B cells, monocytes, and fetal thymus, as well as lung, neuronal, and intestinal tissues (S6 Table). This suggests that it lies in a bivalent or poised regulatory region, meaning that the area usually stays inactive but is ready to turn on when needed, potentially influencing gene expression during

immune activation or inflammation. On the other hand, rs112982286 shows similar regressive marks (H3K27me3) in the myeloid progenitors and kidney, while rs79900659 lies in the regions marked by H3K9me3, a silencing mark, in myeloid cells, adrenal gland and fetal muscle (S6 Table). Together, these variants may affect the regulation of genes involved in immune differentiation, inflammatory responses, and multi-organ involvement, which are central to COVID-19 severity. Importantly, GTEx data revealed that these variants strongly regulate a novel transcript *ENSG00000233633.2* in spleen tissue (p-value=4e-06 to 6e-06; NES~0.95) (S4 Table).

## Discussion

The clinical heterogeneity of COVID-19, ranging from asymptomatic cases to respiratory failure and death, has highlighted the critical need to understand host genetic contributions to disease severity [35]. While several loci have been identified by international consortia in the European populations, the South Asian (SAS) genetic landscape remains poorly characterized, posing a significant gap in our understandingof a pandemic like COVID-19 [36].Our study addresses this critical lacuna by presenting the first GWAS in a genotyped Indian cohort using a dual reference-based imputation approach, leveraging both globally representative (1KGenomes) and Indian population-specific (IndiGen) panels [25,37].

This dual-panel approach highlights two key insights: firstly, the use of population-specific reference panels improves variant yield and fine-mapping resolution, and secondly, novel, biologically relevant signals, especially in the non-coding regions, may be overlooked without ancestry-informed imputation. Following QC and imputation, we obtained 1,133,717 variants using IndiGen, compared to the 729,224 variants with 1KGenomes, reflecting a 55.47% increase in high-confidence variant detection with the Indian population-specific reference panel. Notably, 35.67% of the variants captured using IndiGen were missed when using the 1KGenomes panel, underscoring the value of a population-specific reference panel. Importantly these differences are not merely technical; they translate into tangible biological insights, especially into COVID-19-relevant regulatory mechanisms in the lung and immune cells (**Fig 6**).

From the 1KGenomes imputed dataset, we identified a high-confidence LD block centred on chromosome 2 (rs35575084) within a non-coding genomic region containing the lincRNA *AC007100.1* and the miRNA host gene *MIR4432HG*. This locus encodes *miR-4432*, a regulatory miRNA with emerging relevance in endothelial biology. *miR-4432* suppresses *FGFBP1* (*Fibroblast Growth Factor Binding Protein 1*), a secreted binding protein that has been recently identified as a biomarker for persistent SARS-CoV-2 replication and endothelial injury in ventilated COVID-19 patients [26,27]. *FGFBP1* is expressed in endothelial cells and regulates vascular homeostasis by modulating endothelial sensitivity to vasoactive stimuli, including angiotensin II, through activation of reactive oxygen species and MAP kinase signalling pathways [38,39]. Elevated *FGFBP1* expression has been shown to promote sustained vascular inflammation, endothelial dysfunction, and heightened vasoconstrictor responsiveness in both human and experimental models [26]. These processes overlap with key features of severe COVID-19, including endothelial activation and hyper-inflammatory vascular signalling. Taken together, and in line with prior functional evidence, these observations support a biologically plausible mechanistic link between regulatory variation at the MIR4432HG locus, *FGFBP1*-mediated endothelial dysfunction, and COVID-19 associated vascular inflammation [40].Moreover, *miR-4432* is selectively packaged into extracellular vesicles (EVs) by endothelial cells in a Ribosomal Protein L36 (RPL36)-dependent manner, and such EV-encapsulated *miR-4432* can disrupt perivascular cell adhesion, differentiation, and proliferation [33]. This mechanism mirrors the clinical observations of vascular pathology frequently reported in severe COVID-19, including endothelialitis and microvascular thrombosis [40,41]. Importantly, *miR-4432* has not yet been incorporated into existing miRNA-based severity prediction models, highlighting its novelty and the potential for translational applications. Taken together, this locus illustrates how non-coding variants in vascular-specific pathways may exert system-wide effects and could contribute to the pathophysiology of severe COVID-19. While direct differential expression of AC007100.1 or MIR4432HG in the COVID-19 patient tissues has not yet been reported, the locus was interpreted as a putative regulatory region based on its LD structure and prior functional evidence linking miR-4432 to endothelial biology.

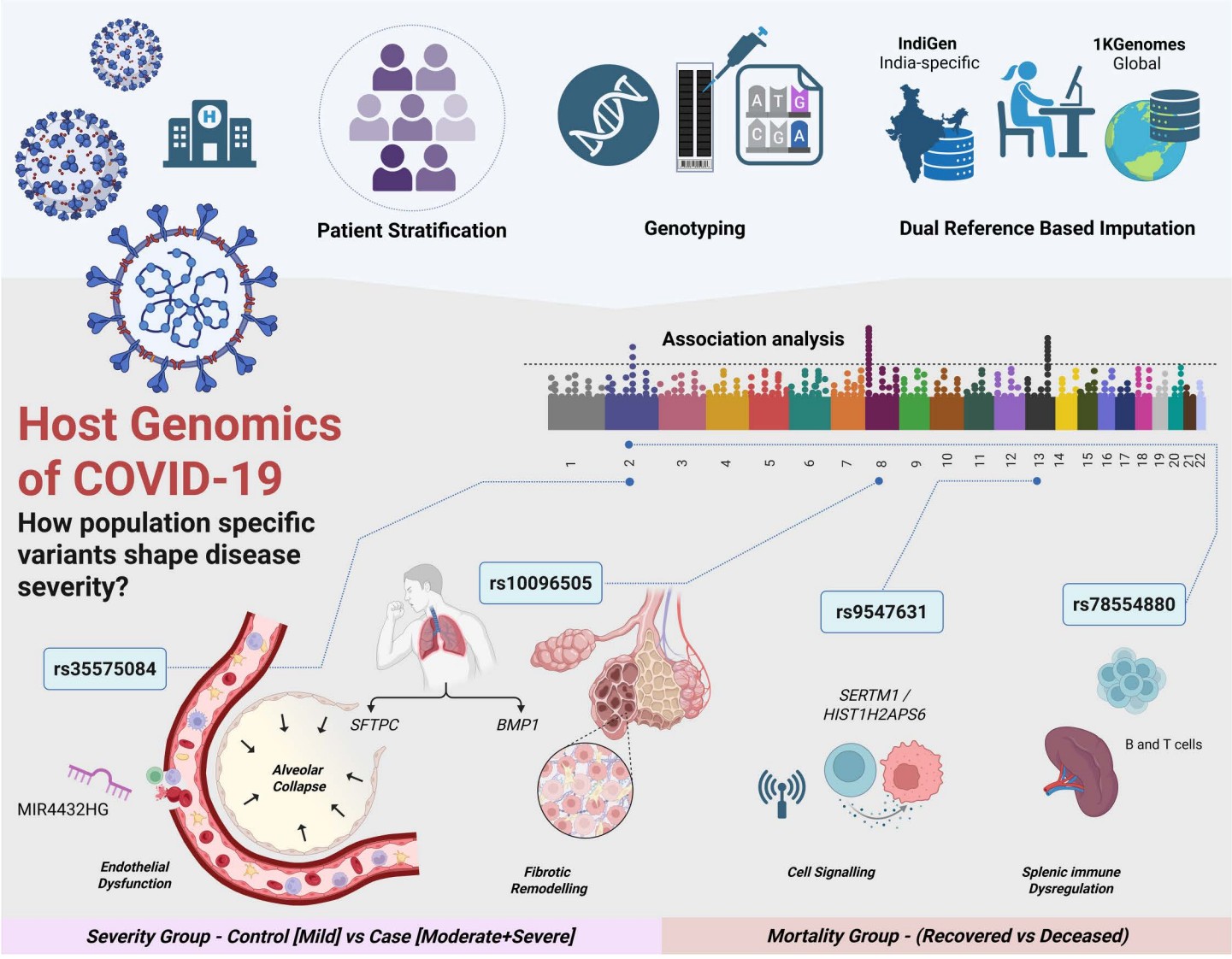

**Fig 6. Summary of the key genomic loci identified from comparative GWAS points towards plausible regulatory mechanisms linked to COVID-19 severity.** The study stratifies patients based on the COVID-19 symptoms and clinical outcomes, and performs genotyping followed by imputation using both a global (1KGenomes) and an indigenous (IndiGen) reference panel. Post-imputation, GWAS identified population-specific variants linked to the COVID-19 severity '*Control (Mild) vs Case (Moderate + Severe)*' and mortality '*Recovered vs Deceased*'. Key loci, such as rs35575084 near *MIR4432HG* linked to the endothelial dysfunction, rs10096505 associated with the alveolar collapse and fibrotic remodelling, rs9547631 potentially implicated in cell signalling, and rs78554880 linked to splenic immune dysfunction, are highlighted. The approach reveals how genetic diversity across populations may shape different host responses to SARS-CoV-2 infection. Created using a licensed version of BioRender.com in the lab.

The LD block surrounding rs35575084 represents a group of tightly linked variants that is consistent with LD patterns commonly observed in the SAS populations. Indian and broader SAS populations are known to show extended LD in certain genomic regions due to shared ancestry and historical admixture. While a direct one-to-one LD comparison for this exact locus in external SAS reference datasets (such as 1000 Genomes) is limited by the sample size and allele frequency differences, the presence of a coherent LD block in our data is expected and consistent with the known population-specific

LD architectures. Importantly, the relevance of this LD block is supported not by LD structure alone, but by functional annotation. The associated variants lie in the non-coding regions and overlapping regulatory chromatin marks across the endothelial, immune, and lung-related tissues, and are located near MIR4432HG, which gives rise to miR-4432. Previous studies have shown that miR-4432 regulates FGFBP1 in endothelial cells, linking this regulatory region to endothelial dysfunction, which is a key pathological feature of severe COVID-19.

In contrast to the vascular regulatory signal described above, the IndiGen panel identified rs10096505 located on chromosome 8, representing a compelling lung epithelial vulnerability signal. This variant functions as a splicing QTL (sQTL) for *SFTPC* as well as an eQTL for *BMP1*, implicating it in alveolar surfactant homeostasis and pulmonary fibrosis, respectively. *SFTPC* encodes Surfactant Protein C for maintaining alveolar surface tension and epithelial integrity [31]. Disrupted splicing may compromise gas exchange and predispose to alveolar collapse, particularly under inflammatory stress [32] aligning with the clinical conditions of hypoxia and respiratory distress in severe COVID-19 cases. The transcriptomic analyses have shown that downregulation of *SFTPC* correlates with high viral load in the lungs. [42], while other *SFTPC* variants such as rs8192340 and rs8192330 have previously been associated with protection from pediatric COVID-19 and altered splicing patterns in adults with high viral load [43,44]. Additionally, *SFTPC* contributes to immune regulation via the JAK/STAT pathway, providing a mechanistic bridge between lung-associated epithelial injury and immune dysregulation [45]. These observations indicate that surfactant dysfunction may represent an upstream contributor rather than solely a downstream consequence of disease severity.

Simultaneously, rs10096505 acts as an eQTL for *BMP1,* a gene involved in the extracellular matrix (ECM) processing and fibrosis. *BMP1* expression is upregulated in COVID-19 lung injury, and genetic studies have linked it to fibrotic diseases such as systemic sclerosis and rheumatoid arthritis [46]. *BMP1* also influences IL-6, a major cytokine inducing hyper-inflammation in severe COVID-19 [47,48]. Thus, rs10096505 may represent a regulatory nexus connecting surfactant biology, endothelial stress, fibrosis and immune dysregulation. When considered together, our findings suggest a triad and coordinated pattern of pulmonary pathophysiology underlying severe COVID-19. To begin with, alveolar dysfunction via disrupted *SFTPC* splicing (rs10096505), second, fibrotic remodelling through upregulated *BMP1* expression (rs10096505) and third, endothelial stress mediated by EV-packed *miR-4432* (rs35575084). These distinct yet convergent regulatory signals operate within the same lung microenvironment, highlighting the importance of tissue-specific, non-coding variation in the COVID-19 severity. It is important to note that while the identified variants did not reach genome-wide significance, they represent suggestive associations which remain exploratory and warrant further validation in larger, ancestry-matched cohorts.

Mortality-specific association analyses identified two suggestive loci. The variant rs9547631 on chromosome 13, situated between *SERTM1* and *HIST1H2APS6,* was detected consistently across both panels. Although functional annotations remain limited, *SERTM,1,* containing a Ser-rich transmembrane (TM) domain, may be involved in cell signalling [49]. The second locus, identified only in IndiGen-imputed data, involves four tightly linked variants near lincRNAs *AC116609.1* and *AC116609.2*. These variants regulate the expression of *ENSG00000233633.2* in spleen tissue, which plays a central role in adaptive immunity. Given that SARS-CoV-2 can directly target the spleen [50], this locus may reflect compromised immune activation in individuals with fatal outcomes.It is important to note that the deceased cohort in this study was limited in size (N = 66), reflecting the availability of real-world clinical samples collected during the first wave of the COVID-19 pandemic. This limited sample size impacts the statistical power, potential susceptibility to false-positive findings, and may inflate effect size estimates. Being conscious of this, accordingly, all mortality-associated signals identified here are interpreted as exploratory and hypothesis-generating rather than definitive genetic risk loci. The primary objective of this analysis is to prioritize ancestry-relevant regulatory regions for further investigation, with the expectation that validation in larger, independent, and ancestry-matched cohorts will be required as such datasets become available.

This study reinforces the importance of incorporating ancestry-specific genomic resources. PCA results showed our cohort clustered predominantly with the SAS super-population, validating the use of Indian-specific

panels for accurate imputation. The IndiGen panel not only yielded more variants but also unveiled India-specific associations, such as rs10096505 and the mortality-linked chromosome 2 block, that were entirely missed using 1KGenomes. This disparity highlights the urgent need to expand population-specific imputation resources and harmonize global pipelines for equitable genomic discovery. From a methodological standpoint, our findings echo a growing consensus: non-coding regulatory variants, particularly eQTLs and sQTLs in the disease-relevant tissues (lungs, spleen, endothelium), are key drivers of complex disease phenotypes. COVID-19 pathophysiology, as demonstrated here, may be as much about regulatory imbalance as it is about viral load and immune hyper-activation.

Importantly, Indian populations are known to exhibit a complex genetic structure shaped by historical admixture between multiple ancestral components, including Ancestral North Indian (ANI), Ancestral South Indian (ASI), and East/Southeast Asian-related ancestries. Large-scale population genetic studies have consistently demonstrated gradients of East Asian-related ancestry within the SAS populations, particularly among groups from North, North-East, and Eastern India [51,52]. Consequently, modest shifts of some Indian individuals towards the EAS cluster in global PCA reflect genuine population substructure rather than population misclassification or technical artefacts. Hence, these individuals were retained in the downstream analysis to avoid over-correction of true population structure.

Alongside host genetic factors, it is also recognized that circulating SARS-CoV-2 variants of concern (VOC) have been reported to differ in their clinical severity and outcomes, highlighting the potential contribution of viral diversity to clinical heterogeneity. In the present study, viral lineage information was not available for stratified analysis as the study was specifically designed to investigate host genetic susceptibility using a uniform clinical classification criterion across patients recruited during a single time window corresponding to the first wave of COVID-19 in India. As the samples were not collected across multiple epidemic phases, variant-driven heterogeneity is expected to be limited. Nevertheless, future studies integrating host genotype data with viral genomic information will be important to further disentangle host and virus driving contributors to COVID-19 severity [53–55].

While our study offers important insights into the host genomics of COVID-19 in the Indian population, several limitations still require attention. Although this represents one of the largest genotyped Indian cohort to date, but the sample size remains modest compared to the global consortia, which may limit power to detect associations with small effect sizes. There are some associations, such as non-coding transcripts (e.g., *ENSG00000233633.2*) remain hypothetical due to the absence of functional validation. Although we applied rigorous quality control, population substructures within the Indian population could still confound associations. Future directions include targeting sequencing around suggestive SNPs to refine genetic associations signals, as well as functional validation of key loci using CRISPR editing, single-cell multi-omics and epigenomic profiling. Replication in larger, independent Indian cohorts will be essential to strengthen and validate these findings.

## Conclusion

Our study presents the first dual-imputed GWAS of COVID-19 in a genotyped Indian cohort, revealing novel, biologically plausible, and tissue-specific regulatory mechanisms of disease severity and mortality. Through ancestry-aware imputation, we identified suggestive, non-coding loci linked to endothelial, epithelial, and immune processes loci that would remain undetected in traditional, Eurocentric reference frameworks. India's unique population structure is shaped by endogamy, tribal diversity, and ancient migrations, necessitating genomic tools tailored to its genetic context. Our findings not only validate the utility of the IndiGen panel but also advocate for its broader integration into clinical genomics, imputation pipelines, and public health decision making. Ultimately, our work contributes to the growing body of evidence that genomic equity achieved through representative reference panels and population-specific studies is essential for understanding complex disease biology and ensuring precision medicine reaches all populations.

## Materials and methods

### Ethics statement

The study followed the Declaration of Helsinki guidelines. It was approved by the institutional ethics committee of the Council of Scientific & Industrial Research-Institute of Genomics and Integrative Biology, Delhi, India (Ref No: CSIR-IGIB/IHEC/2020–21/01). The patients provided written informed consent to participate in this study.

### Study population and sample collection

Between May and September 2020, during the first wave of the COVID-19 pandemic, blood samples were collected from patients with SARS-CoV-2 infection confirmed by Real-Time reverse transcription (RT-PCR) at the Max Super Speciality Hospital, Saket, in Delhi. Their diagnosis was confirmed through routine clinical procedures at the hospital. The samples were later sent to our laboratory at the CSIR-Institute of Genomics and Integrative Biology (IGIB) for the subsequent experiments. A total of 677 patients were selected for the study cohort. For each participant, clinical and demographic metadata (e.g., age, sex, comorbidities, symptoms, oxygen saturation, need for supplemental oxygen support) were meticulously recorded at the time of hospital admission. All data were de-identified, and the study was conducted following ethical approvals and informed consent.

### Sample processing to quality-controlled genotypic data

Genomic DNA was extracted from the whole blood using the MasterPure DNA Purification Kit for Blood, Version II (Lucigen; Cat. No. MB711400). Extractions were performed according to the manufacturer's protocol. DNA concentration and purity were measured using a NanoDrop spectrophotometer (Thermo Fisher Scientific), and only samples with A260/280 ratios between 1.8 and 2.0 and concentrations ≥50 ng/μL were selected for downstream processing. These thresholds were selected to ensure high-quality DNA for reliable genotyping. Genotyping was performed using the Infinium High-Throughput Screening (HTS) Assay (Illumina, San Diego, CA, Cat. No. 20030770) on the Infinium Global Screening Array (GSA) v3.0 BeadChip, targeting 6,54,027 markers genome-wide. Briefly, 200 ng of DNA per sample was denatured, isothermally amplified, enzymatically fragmented, and hybridised to the BeadChip overnight. Post-hybridisation, single-base extension, and staining were performed, and chips were scanned. Intensity data files (.IDAT) were generated for downstream analysis using the iScan imaging system.

The initial level of sample quality control (QC) was conducted using GenomeStudio v2.0, excluding samples with call rate (CR) < 95%, and GenTrain score < 0.7, resulting in a total of 669 high-quality samples with 6,17,090 variants. QC thresholds were chosen in accordance with standard GWAS best practices. The clinical metadata has been summarized in S1 Table. The genotypic data were exported from GenomeStudio in PLINK format with forward strand orientation to ensure compatibility with downstream imputation pipelines. Subsequent variant- and sample-level QC steps were performed using PLINKv1.9, removing individuals and SNPs with > 5% missingness, sex discrepancies, non-autosomal SNPs, variants deviating from the Hardy-Weinberg Equilibrium (p-value < 1e-06), and those with minor allele frequency (MAF) < 0.05. LD pruning ($r^2$ > 0.2) was applied to obtain independent markers for the population structure analysis. Finally, relatedness was checked to remove cryptic or known relatives. Pairs of individuals with pi-hat > 0.2 were identified, and samples were removed based on this metric and in combination with clinical metadata review, ensuring a cohort of 610 unrelated individuals with 104,987 variants was used for the association analysis.

### Clinical phenotype classification

To investigate host genetic variants associated with progression to severe forms of COVID-19, we employed a binary case-control framework for the 610 individuals, adhering to the clinical guidelines provided by the Indian Council of

Medical Research (ICMR). The patient subgroups were established based on the key respiratory parameters, including oxygen saturation (SpO$_2$), respiratory rate, and respiratory support, as well as Intensive Care Unit (ICU) admission.

### Severity-based classification

**Control group for association testing (_N = 254, Mild cases_).** The control group included patients with SpO$_2 \geq 94\%$ on room air, a respiratory rate of ≤ 23 breaths/min, and no clinical signs of respiratory distress. These patients did not require oxygen or ventilator support, were not admitted to the ICU, only experienced mild symptoms (e.g., fever, sore throat, malaise), and were stated to have recovered without escalation of care. These patients showcased relative resistance against the disease progression.

**Case group for association testing (_N = 356, Moderate + Severe cases_).** The case group included patients with SpO$_2 < 94\%$, elevated respiratory rate (≥24 breaths/min), signs of respiratory distress (e.g., shortness of breath) or those requiring oxygen therapy, non-invasive or invasive ventilator support, or who were admitted to the ICU. _Moderate cases_ (N = 193) were defined as patients with SpO$_2$ levels between 90–93%, a decrease in their respiratory rate from baseline, having a respiratory rate of between 24–29 breaths/minute, showing respiratory symptoms, which were noted as shortness of breath, and often required non-invasive oxygen support. ICU admission was necessary in some cases. _Severe cases_ (N = 163) were defined as patients who showed their SpO$_2$ levels at ≤ 90%, respiratory rate ≥ 30 per minute, and the need for supplemental oxygen and ventilator support. Severe cases were treated in the ICU and were also typically at higher risk of mortality. Given the clinical overlap in disease trajectories, treatment needs, and respiratory support criteria, moderate and severe patients were merged into a single **"Case"** group for association analysis.

### Outcome-based classification

In parallel to severity-based classification, the patients were also classified according to the disease outcome for capturing definitive clinical endpoints and identifying factors associated with survival or death. Patients who were discharged after recovery were categorized as the "Recovered" group (N = 524), while those who succumbed during hospitalization due to COVID-19-related complications were assigned to the "Deceased" group (N = 66). Patients with unavailable outcome data or those who left against medical advice (LAMA) were excluded from the outcome-based analysis.

### Phasing, genome build harmonization and liftover

The 1KGenomes (Phase III) reference panel was downloaded from (https://www.cog-genomics.org/plink/2.0/resources#phase3_1kg) [56], comprising 2504 individuals from 26 global populations, and originally containing over 84 million variants mapped to the GRCh38/hg38 reference genome [57]. The IndiGen reference panel developed by CSIR-IGIB comprises 1029 healthy Indian genomes from diverse ethnolinguistic backgrounds and approximately 55 million biallelic variants, also aligned to the hg38 build [25]. Both panels underwent the same QC filters as our COVID-19 genotypic data, including missingness filtering, Hardy-Weinberg Equilibrium (HWE), and minor allele frequency (MAF) thresholds, ensuring uniform data quality and compatibility for imputation. As both reference panels were initially unphased, they were phased using SHAPEIT2 with default settings [58]. Our COVID-19 data, originally aligned to the GRCh37 (hg19), was converted to hg38 using the 'rtracklayer' package in R to ensure compatibility with the reference panels [59]. After harmonization, 104,946 variants from 610 unrelated individuals were retained for imputation and association analysis.

### Population structure and outlier detection

Population structure was assessed using Principal Component Analysis (PCA) after genotype-level quality control. To assess global ancestry composition, we performed a two-dimensional (2D) PCA by projecting study samples directly onto

reference populations from the 1000 Genomes Project, enabling direct comparison with major continental populations (Fig 1B-i). This analysis demonstrated predominant clustering of the cohort with the South Asian (SAS) super-population. To enhance resolution and identify potential extreme ancestry deviations, a three-dimensional PCA (3D) was also examined using a Z-score threshold ≥ 6, which did not reveal any additional population outliers (S1 Fig). In addition, a within-cohort PCA restricted to the GSA dataset was performed to identify potential technical or extreme population outliers independent of the global reference structure. This analysis identified two clear outliers, which were removed before imputation and downstream association analyses (Fig 1B-ii and 1B-iii). The top 10 principal components (PC1-PC10) derived from the PCA were included as covariates in all the association analyses

## Genotype imputation

Following population structure analysis and outlier removal, the genotype data from 608 unrelated individuals were prepared for imputation. To increase genome-wide coverage, imputation was performed using Beaglev5.5, a widely accepted imputation tool based on the localised haplotype clustering [60]. Imputation was carried out separately using both reference panels (1KGenomes and IndiGen) and yielded probabilistic genotype dosages for each untyped variant. To retain only high-confidence variants, we filtered the imputed data based on the dosage R-squared ($DR^2$) metric, which reflects the squared correlation between the true genotypes and imputed genotype dosages [61]. A threshold of $DR^2 \geq 0.8$ was applied to retain only well-imputed variants. Following imputation and filtering, 729,224 variants were obtained using 1KGenomes, whereas 1,133,717 variants were obtained using IndiGen.

## Association testing

Association testing was performed using logistic regression models in PLINKv1.9, adjusting for age, sex, and comorbidities (hypertension, diabetes, cardiovascular and respiratory diseases and cancer) as covariates. To explicitly control for population stratification and ancestry differences between the comparison groups, the top 10 principal components (PC1-PC10) derived from PCA were included as covariates in all the models. Covariates were selected based on known clinical risk factors for COVID-19 severity. An additive genetic model was applied to assess the association between imputed variants and COVID-19 severity. Analyses were performed separately for: i) *Control (Mild) vs Case (Moderate + Severe)* groups, and ii) *Recovered vs Deceased* patients, to evaluate the impact of host genetics on disease progression and outcomes. The same covariate framework (age, sex, comorbidities, and PC1-PC10) was applied consistently across both the severity and deceased analyses and across both the imputation panels. Genome-wide significance was defined as $p < 5 \times 10^{-8}$, and variants with $p < 1 \times 10^{-5}$ were considered suggestive. Association testing was carried out separately for data imputed with the 1KGenomes and IndiGen reference panels, allowing for comparative identification of global and population-specific variants.

## LD and functional annotations

To further prioritize biologically relevant loci, suggestive SNPs were analysed for linkage disequilibrium (LD) with neighboring variants within a ± 1Mbp genomic window. LD calculations were performed using a window-based approach in PLINKv1.9, and SNP pairs were categorized into strong ($r^2 \geq 0.8$), moderate ($0.5 \leq r^2 < 0.8$), and weak LD ($r^2 < 0.5$). SNPs in strong LD were selected for further analysis to identify candidate genes potentially affected by the variant signals. To map these SNPs to nearby genes, the SNPNexus web-based tool (https://www.snp-nexus.org/v4/) [62–64] was used, which provides gene annotations based on both upstream (UST) and downstream (DST) proximity to the variant. The biological roles of these candidate genes were further interpreted through manual curation of existing literature and databases, focusing on the immune response and lung-specific pathways relevant to the SARS-CoV-2 infections. For expression-based functional insights, expression quantitative loci (eQTL) analysis was performed for the suggestive SNPs

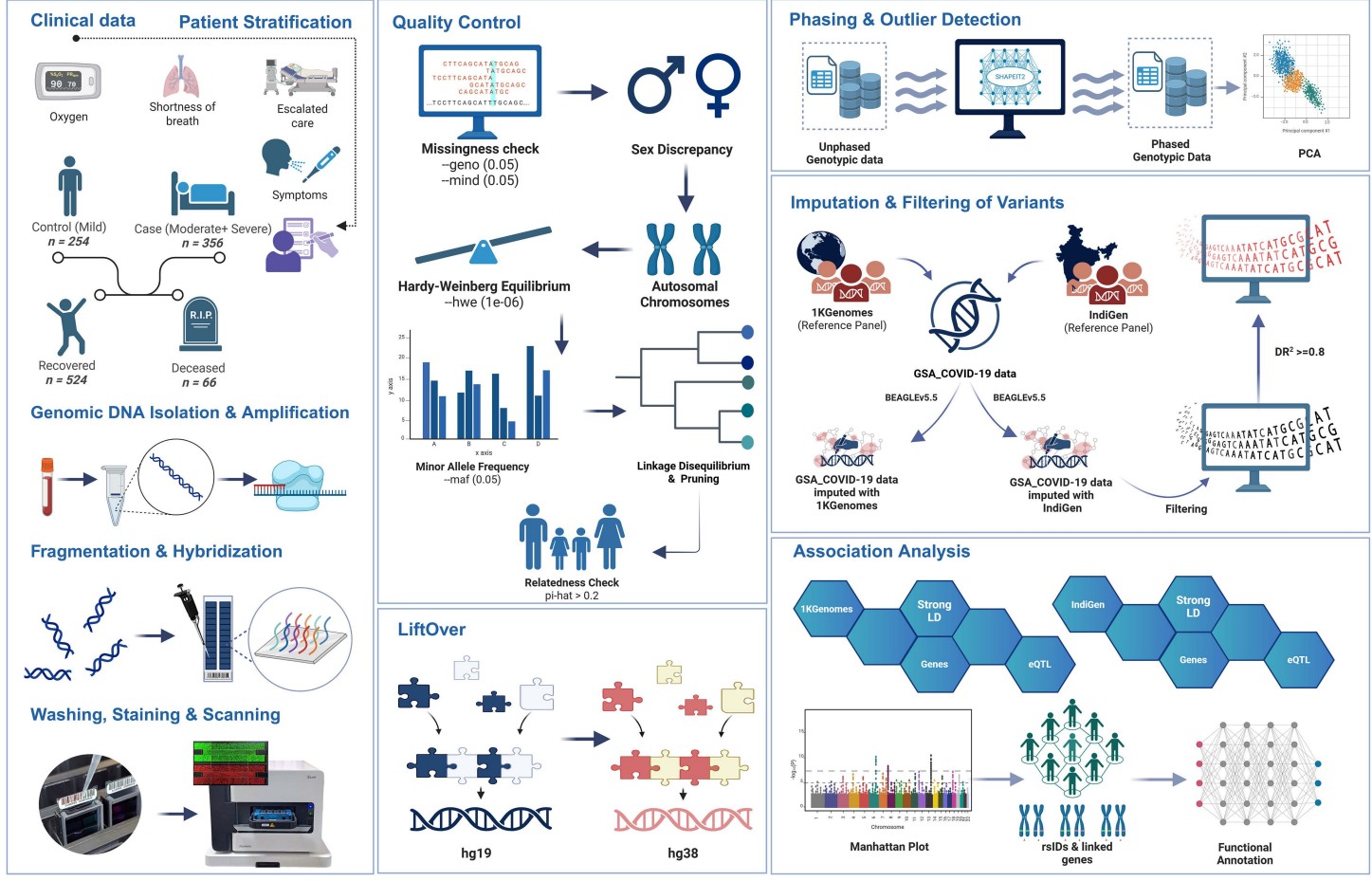

**Fig 7. Schematic workflow illustrating comparative GWAS with 1KGenomes and IndiGen reference panels.** The pipeline begins with patient stratification based on the COVID-19 symptoms, followed by genomic DNA extraction, amplification, and genotyping. Post-genotyping quality control includes checks for missingness, sex discrepancies, HWE, relatedness, and MAF. Unphased genotype data are then phased, and outlier samples are identified via PCA. Imputation is performed separately using the 1KGenomes and IndiGen reference panels via BEAGLE5.5, and filtered based on imputation accuracy ($DR^2 \geq 0.8$). Downstream association analyses identify population-specific loci and variants in strong LD, which are further functionally annotated using eQTL and gene mapping tools. This approach enables cross-panel comparisons of the genetic architecture associated with COVID-19 severity and outcomes. Created using licensed version of BioRender.com in the lab.

using the GTEx portal (https://gtexportal.org/home/). Tissue-specific expression changes (UST/DST) of nearby genes were examined, particularly in the COVID-19-relevant tissues such as the lungs, spleen, and blood cells**. The methodology to perform a comparative GWAS has been outlined in **Fig 7**.

## Supporting information

**S1 Fig. Population stratification visualized by 3D PCA.** 3D PCA revealed that none of the COVID-19 samples exhibited a Z-score greater than 6, and the samples clustered with the South Asian population, indicating no major outliers. (PDF)

**S1 Table. Clinical metadata of 669 COVID-19 patients.**

(XLSX)

**S2 Table. SNPs in LD with the suggestive COVID-19 variants.**
(XLSX)

**S3 Table. SNPnexus-based annotation of COVID-19 associated variants.**
(XLSX)

**S4 Table. Functional annotations based on GTEx expression data.**
(XLSX)

**S5 Table. SNPs near +/-1Mbp region of suggestive SNPs using 1KGenomes and IndiGen.**
(XLSX)

**S6 Table. Mapping of chromatin features using RoadMap Epigenome data.**
(XLSX)

## Acknowledgments

The authors would like to acknowledge all the recruited COVID-19 patients for their cooperation and consent. We thank Dr. Aradhita Baral and Dr. Bharti for their support in facilitation as research managers, including coordination with funders, hospital partnerships, and administrative processes. We are grateful to Anil and Nisha for their contributions toward the timely transport and management of COVID-19 samples. We also thank Aparna Swaminathan, Jyoti Soni and Dr. Thierry Habyarimana for their valuable contributions in the experimental work. We further acknowledge the use of the IndiGenomes reference panel in this study, available from NCBI under Submission ID: SUB8153462, and accessible via the IndiGenomes database at http://clingen.igib.res.in/indigen/.

## Author contributions

**Conceptualization:** Uzma Shamim, Rajesh Pandey.

**Data curation:** Aastha Kaushik, Ramakant Mohite, Ranjeet Maurya, Uzma Shamim.

**Formal analysis:** Aastha Kaushik, Ranjeet Maurya.

**Funding acquisition:** Rajesh Pandey.

**Investigation:** Ramakant Mohite.

**Methodology:** Aastha Kaushik, Ramakant Mohite, Ranjeet Maurya.

**Project administration:** Rajesh Pandey.

**Resources:** Bansidhar Tarai, Sandeep Budhiraja, Rajesh Pandey.

**Software:** Aastha Kaushik, Ranjeet Maurya.

**Supervision:** Uzma Shamim, Rajesh Pandey.

**Visualization:** Aastha Kaushik, Ramakant Mohite, Ranjeet Maurya.

**Writing – original draft:** Aastha Kaushik, Ramakant Mohite.

**Writing – review & editing:** Aastha Kaushik, Ramakant Mohite, Ranjeet Maurya, Uzma Shamim, Rajesh Pandey.

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
