## [Decision Letter · Decision Letter 0]

10 Dec 2025

Response to Reviewers
Revised Manuscript with Track Changes
Manuscript

Shaden Kamhawi

co-Editor-in-Chief

Paul Brindley

co-Editor-in-Chief

**Journal Requirements:**

At this stage, the following Authors/Authors require contributions: Aastha Kaushik, Ramakant Mohite, Ranjeet Maurya, Bansidhar Tarai, Sandeep Budhiraja, Uzma Shamim, and Rajesh Pandey. Please ensure that the full contributions of each author are acknowledged in the "Add/Edit/Remove Authors" section of our submission form.

2) Some material included in your submission may be copyrighted. According to PLOSu2019s copyright policy, authors who use figures or other material (e.g., graphics, clipart, maps) from another author or copyright holder must demonstrate or obtain permission to publish this material under the Creative Commons Attribution 4.0 International (CC BY 4.0) License used by PLOS journals. Please closely review the details of PLOSu2019s copyright requirements here: PLOS Licenses and Copyright. If you need to request permissions from a copyright holder, you may use PLOS's Copyright Content Permission form.

Potential Copyright Issues:

- Figures 5 and 6.. Please confirm whether you drew the images / clip-art within the figure panels by hand. If you did not draw the images, please provide (a) a link to the source of the images or icons and their license / terms of use; or (b) written permission from the copyright holder to publish the images or icons under our CC BY 4.0 license. Alternatively, you may replace the images with open source alternatives. See these open source resources you may use to replace images / clip-art:

**Reviewers' comments:**

**Key Review Criteria Required for Acceptance?**

**Methods**

-Are the objectives of the study clearly articulated with a clear testable hypothesis stated?

-Is the study design appropriate to address the stated objectives?

-Is the population clearly described and appropriate for the hypothesis being tested?

-Is the sample size sufficient to ensure adequate power to address the hypothesis being tested?

-Were correct statistical analysis used to support conclusions?

-Are there concerns about ethical or regulatory requirements being met?

Reviewer #1: (No Response)

Reviewer #2: The study design appears appropriate for addressing the stated objectives, but certain methodological descriptions lack clarity, particularly in the handling of population structure and sample retention.

The population is broadly described and seems appropriate for the hypotheses being tested; however, clearer documentation of ancestry determination, the treatment of population outliers, and the rationale for retaining or excluding individuals with partial EAS overlap is needed.

The mortality sample size (N = 66) is small and raises concerns regarding statistical power, false-positive risk, and potential effect size inflation. These limitations should be explicitly acknowledged by the authors.

**Results**

-Does the analysis presented match the analysis plan?

-Are the results clearly and completely presented?

-Are the figures (Tables, Images) of sufficient quality for clarity?

Reviewer #1: (No Response)

Reviewer #2: The manuscript presents interesting exploratory findings regarding mortality-associated genomic variants, including a robust chromosome 13 signal and a potentially novel regulatory region on chromosome 2.Results are presented clearly in narrative form, but they are incomplete without the key statistical parameters needed to assess robustness

**Conclusions**

-Are the conclusions supported by the data presented?

-Are the limitations of analysis clearly described?

-Do the authors discuss how these data can be helpful to advance our understanding of the topic under study?

-Is public health relevance addressed?

Reviewer #1: (No Response)

Reviewer #2: The authors must more clearly describe the limitations, especially regarding small case numbers, potential confounding by ancestry, and the exploratory nature of the findings.

**Editorial and Data Presentation Modifications?**

Reviewer #1: (No Response)

Reviewer #2: Minor editorial adjustments would improve clarity and readability throughout the manuscript. In particular, clearer reporting of key statistical metrics (e.g., p-values, effect sizes, confidence intervals), more explicit descriptions of population structure analyses, and improved labeling or annotation of figures/tables would strengthen data presentation. These refinements are minor but would enhance transparency and interpretability for readers.

**Summary and General Comments**

Reviewer #1: This study investigates the genetic differences among people in India that may affect how severely COVID-19 impacts them and their chances of survival. The authors suggest that key indigenous genetic signals influencing disease severity and outcomes might be overlooked if imputation is based on a reference panel that underrepresents the studied population. The study found some loci were shared between the IndiGen and 1KGenomes panels, while others were unique to the Indian population. The manuscript is well written, with data analyzed systematically and thoroughly.

Minor points/queries:

1. Line 492: “…samples were collected from patients with confirmed SARS-CoV-2 infection...” Please specify how the infections were confirmed.

2. To ease readers' understanding, consider using “Mild vs Severe” instead of “Control vs Case”. Justification: “Control” generally means no disease or infection. By using “Mild vs Severe”, readers are reminded throughout the manuscript that the subjects were all confirmed infected with SARS-CoV-2 and exhibited clinical symptoms.

3. Consider replacing “Recovered vs Mortality” with “Recovered vs Deceased” instead in the figures, tables, and text.

4. Were the key respiratory parameters listed in the supplementary table and used in determining the patient subgroups obtained at the point of admission?

5. Different SARS-CoV-2 variants are also known to affect disease severity and clinical outcomes. It is suggested that a brief discussion of this is included in the manuscript.

Reviewer #2: The case–control design comparing recovered individuals to mortality cases is appropriate for detecting potential risk loci. However, the extremely limited mortality sample size (N=66) substantially restricts the statistical power and increases the possibility of unstable estimates or false positives. This limitation should be more clearly highlighted in the methods and discussion.

PLOS authors have the option to publish the peer review history of their article (what does this mean? ). If published, this will include your full peer review and any attached files.

**Do you want your identity to be public for this peer review?** For information about this choice, including consent withdrawal, please see our Privacy Policy .

Reviewer #1: No

Reviewer #2: **Yes:** Alazar Amare amare

**Figure resubmission:**

**Reproducibility:** To enhance the reproducibility of your results, we recommend that authors of applicable studies deposit laboratory protocols in protocols.io, where a protocol can be assigned its own identifier (DOI) such that it can be cited independently in the future. Additionally, PLOS ONE offers an option to publish peer-reviewed clinical study protocols. Read more information on sharing protocols at https://plos.org/protocols?utm_medium=editorial-email&utm_source=authorletters&utm_campaign=protocols

---

## [Decision Letter · Decision Letter 1]

9 Feb 2026

Dear Dr. Pandey,

We are pleased to inform you that your manuscript 'Comparative GWAS using global and Indian Reference Panels reveals non-coding drivers of COVID-19 severity and mortality' has been provisionally accepted for publication in PLOS Neglected Tropical Diseases.

Best regards,

Max Carlos Ramírez-Soto, BSc, MPH, PhD, FRSPH, FECMM

Academic Editor

David Safronetz

Section Editor

Shaden Kamhawi

co-Editor-in-Chief

Paul Brindley

co-Editor-in-Chief

Reviewer's Responses to Questions

**Key Review Criteria Required for Acceptance?**

**Methods**

-Are the objectives of the study clearly articulated with a clear testable hypothesis stated?

-Is the study design appropriate to address the stated objectives?

-Is the population clearly described and appropriate for the hypothesis being tested?

-Is the sample size sufficient to ensure adequate power to address the hypothesis being tested?

-Were correct statistical analysis used to support conclusions?

-Are there concerns about ethical or regulatory requirements being met?

Reviewer #1: (No Response)

Reviewer #2: (No Response)

**Results**

-Does the analysis presented match the analysis plan?

-Are the results clearly and completely presented?

-Are the figures (Tables, Images) of sufficient quality for clarity?

Reviewer #1: (No Response)

Reviewer #2: (No Response)

**Conclusions**

-Are the conclusions supported by the data presented?

-Are the limitations of analysis clearly described?

-Do the authors discuss how these data can be helpful to advance our understanding of the topic under study?

-Is public health relevance addressed?

Reviewer #1: (No Response)

Reviewer #2: (No Response)

**Editorial and Data Presentation Modifications?**

Reviewer #1: (No Response)

Reviewer #2: (No Response)

**Summary and General Comments**

Reviewer #1: The authors’ revised manuscript and the accompanying point-by-point responses to the reviewers’ comments have been reviewed, and they have adequately addressed all the comments/points raised in the previous review round. The revisions have improved the clarity, structure, and scientific rigor (e.g. inclusion of p-values as per Reviewer 2's comments) of the manuscript.

I have no further comments to raise at this stage. The manuscript is now suitable for publication in its current form.

Reviewer #2: The manuscript presents a relevant and interesting study and is generally well written. The objectives are clear, and the findings contribute to the existing body of knowledge. However, a few minor issues should be addressed to further improve clarity, consistency, and presentation. These revisions do not affect the overall quality or conclusions of the study but will enhance its readability and rigor.

Clarify a few statements in the introduction to improve coherence and flow.

Provide minor methodological details or justifications where needed for clarity.

Improve the presentation of tables/figures by ensuring consistent formatting and clear captions.

Expand brief explanations in the discussion to better connect results with existing literature.

Perform minor language and grammatical revisions throughout the manuscript

PLOS authors have the option to publish the peer review history of their article (what does this mean? ). If published, this will include your full peer review and any attached files.

**Do you want your identity to be public for this peer review?** For information about this choice, including consent withdrawal, please see our Privacy Policy .

Reviewer #1: No

Reviewer #2: **Yes:** Alazar Amare Amdiyee

---

## [Editor Report · Acceptance letter]

Dear Dr. Pandey,

We are delighted to inform you that your manuscript, "Comparative GWAS using global and Indian Reference Panels reveals non-coding drivers of COVID-19 severity and mortality," has been formally accepted for publication in PLOS Neglected Tropical Diseases.

Best regards,

Shaden Kamhawi

co-Editor-in-Chief

Paul Brindley

co-Editor-in-Chief
